# Aptamer-Based Lateral Flow Assays: Current Trends in Clinical Diagnostic Rapid Tests

**DOI:** 10.3390/ph15010090

**Published:** 2022-01-13

**Authors:** Marjan Majdinasab, Mihaela Badea, Jean Louis Marty

**Affiliations:** 1Department of Food Science & Technology, School of Agriculture, Shiraz University, Shiraz 71441-65186, Iran; majdinasab@shirazu.ac.ir; 2Faculty of Medicine, Transilvania University of Brasov, Nicolae Balcescu St., No. 56, 500019 Brasov, Romania; mihaela.badea@unitbv.ro; 3Universite de Perpignan Via Domitia, 52 Avenue Paul Alduy, CEDEX, 66860 Perpignan, France

**Keywords:** aptamer, lateral flow assay, clinical, biomarker, rapid test

## Abstract

The lateral flow assay (LFA) is an extensively used paper-based platform for the rapid and on-site detection of different analytes. The method is user-friendly with no need for sophisticated operation and only includes adding sample. Generally, antibodies are employed as the biorecognition elements in the LFA. However, antibodies possess several disadvantages including poor stability, high batch-to-batch variation, long development time, high price and need for ethical approval and cold chain. Because of these limitations, aptamers screened by an in vitro process can be a good alternative to antibodies as biorecognition molecules in the LFA. In recent years, aptamer-based LFAs have been investigated for the detection of different analytes in point-of-care diagnostics. In this review, we summarize the applications of aptamer technology in LFAs in clinical diagnostic rapid tests for the detection of biomarkers, microbial analytes, hormones and antibiotics. Performance, advantages and drawbacks of the developed assays are also discussed.

## 1. Introduction

Lateral flow assays (LFAs), also known as immunochromatographic assays (ICAs) are paper-based assays for the detection of a variety of analytes in different matrixes, where the liquid sample or sample extract is dropped on the test strip and the results are indicated within a few minutes [1]. LFAs are a kind of point of care (POC) testing method that were first introduced as a urine-based pregnancy test by Unipath (a spin out from Unilever at Colworth in the UK) in 1984. They soon became very popular as analytical platforms worldwide such that hundreds of LFAs have been reported and commercialized so far. Ease of production, low production cost, low price, immediate results, high sensitivity and selectivity, and ease of use of LFAs have resulted in the expansion of their uses to different fields in which rapid tests are required. According to these features, they are extensively used in hospitals, health centers, clinical laboratories, and physician’s offices, as well as for quality control, food safety assessment and environmental health [1,2,3,4]. Moreover, LFA-based tests have been very well accepted by users and regulatory authorities.

Application of LFAs in medicine is highly versatile where they can be used for diagnosis, screening, prognosis, monitoring, and surveillance. Unlike conventional methods, they can be applied outside of laboratories and even by patients themselves for home use [5]. Furthermore, due to the long shelf life and no need for special conditions such as using the refrigerator for storage, LFAs are well matched for use in developing countries, small outpatient care centers, remote areas and battlefields [1].

In expressing the special importance of LFA tests, we can mention their application in controlling the global coronavirus disease 2019 (COVID-19) pandemic caused by the novel coronavirus (SARS-CoV-2). In this pandemic, diagnostic testing at the POC was extensively accepted as part of the post restriction COVID-19 control strategy. The LFA as a popular POC diagnostic platform played an important role in controlling the COVID-19 pandemic in industrialized countries and resource-limited settings.

## 2. Design and Principle of Lateral Flow Assays

A LFA is combination of a chromatographic system (separation of the constituents of a mixture based on differences in their motion along the membrane) and a biochemical reaction (between antibody–antigen or nucleic acid–target molecule) [6]. Both antibodies and nucleic acids can be used as biorecognition elements for developing a LFA. The LFA is referred to as a lateral flow (immuno)assay when antibodies are employed as biorecognition element.

During the detection process, the liquid sample or its extract containing target analyte moves across the membrane via capillary force [7]. A typical LFA strip includes four overlapping membranes that have been pasted on an adhesive backing card for better stability and handling. The four membranes that make a single strip of a LFA include: (1) A sample pad for the sample fluid dropping. In some cases, the sample pad (usually composed of cellulose acetate or glass fiber) is impregnated with buffer salts and surfactants which facilitate the movement and reaction of the sample across the strip. Another function of the sample pad in some assays is to perform sample pretreatment such as separation of red blood cells from the sample matrix. (2) Conjugate pad: the area on which the bio-recognition element conjugated to a colored or fluorescent particle is immobilized. (3) Reaction membrane (generally nitrocellulose membrane), on which the primary bio-recognition element is immobilized to provide a test line where the affinity assay between target and DNA probe or between antigen and antibody occurs. (4) Absorbent pad, which acts like a wick and collects the excess sample fluid (Figure 1a) [6,8]. When the sample containing the analyte is dropped on the sample pad, it moves via capillary force towards the conjugate pad where the target analyte binds to the labeled antibody. Then, the analyte–antibody complex flows along the strip into the nitrocellulose membrane where the complex is captured by the immobilized antibody (or antigen–protein conjugate) on the test line. The excess unbound analyte–antibody complex is captured by antibody immobilized on the control line. Detection of the analyte results in a colorimetric or fluorescence signal on the test line, while a signal on the control line indicates the proper liquid flow along the strip. Finally, the excess liquids are collected by the absorbent pad. The generated signal on the test and control lines can be observed by naked eye or read using a dedicated reader.

There are two formats of antibody-based LFAs: (1) sandwich format; (2) competitive format (Figure 1b). The sandwich format is used for large analytes with multiple antigenic sites such as bacteria, hormones (e.g., human chorionic gonadotropin (hCG)), and some biomarkers. In this format, a primary antibody towards the antigen is sprayed on the test line. A secondary antibody against another epitope of the same antigen is labeled with nanoparticles and is immobilized on the conjugate pad. A third species-specific anti-immunoglobulin antibody is sprayed on the control line and can interact with the secondary antibody. In fact, in the sandwich format, the target is immobilized between two complementary antibodies (i.e., primary and secondary antibodies) on the test line. In this format, signal generation in the test line indicates a positive result [1,6]. 

For evaluation of small molecules with low molecular weight and only a single antigenic determinant, the competitive format is used. Because small molecules cannot simultaneously bind to two antibodies, usually only two different antibodies are used in this format. There are two types of competitive format. In the first type (Figure 1b-1), one antibody that is specific with the antigen is conjugated to labels and immobilized on the conjugate pad, and another antibody that is a species-specific anti-immunoglobulin is sprayed on the control line and reacts with reagent particles. In this format, the antigen carrier molecule (generally bovine serum albumin (BSA)) conjugate is dried on the test line on the nitrocellulose membrane. In this case, a competitive reaction takes place between analyte in the sample and analyte-BSA in the test line for binding to the labeled antibody. Therefore, a positive result is provided by the lack of signal in the test line [1,6,9,10]. In both formats, for the detection of multiple analytes, the number of test lines can be increased. In the second type (Figure 1b-2), the specific antibody against antigen is immobilized on the test line and the immobilized labeled antigen (labeled analyte-BSA) on the conjugate pad is used for detection. Therefore, a competitive reaction takes place between analyte in the sample and labeled antigen in the test line for binding to the specific antibody on the test line. The results are similar to the first competitive format. 

LFA can use different materials to label antibody such as latex beads [11], carbon nanoparticles [12], colloidal silver [13], dye-loaded liposomes [14], gold nanoparticles (AuNPs) [7], europium chelate-loaded silica nanoparticles [15], quantum dots [16] and upconverting phosphors [17]. Each label is able to create different characteristics in the diagnostic platform, but the main purpose in using different labels is to try to make a more sensitive assay. However, among them, AuNPs are still the most widely used and the most popular label due to their unique optical properties, giving them the ability to generate an observable colorimetric signal by naked eye, high stability in liquid or dried form, and ability to easily conjugate with biological materials.

**Figure 1 pharmaceuticals-15-00090-f001:**
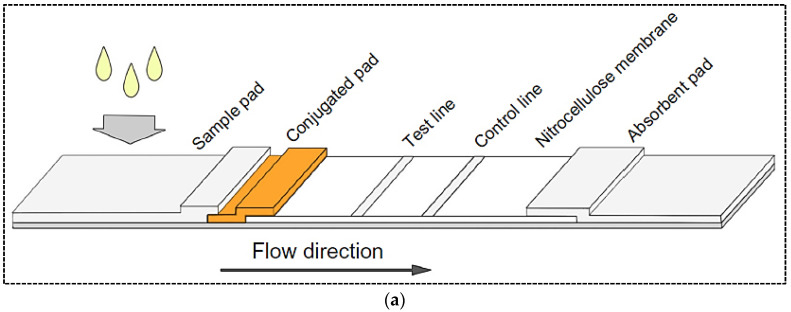
(**a**) Basic structure of a lateral flow test strip; (**b**) Principle of two common formats of LFA including sandwich and competitive formats. Reprinted with permission from [18,19], respectively.

Moreover, different kinds of recognition elements including antibodies, aptamers and molecular beacons can be used for developing a LFA. Although antibodies are the first and most common biorecognition element used in LFA, in recent years, aptamers have also been able to serve as a viable alternative to antibodies in many diagnostic methods, including LFA, due to their potential benefits. The application of aptamers in LFA and the detection principle of aptamer-based LFAs are described in detail below. 

## 3. Replacing Aptamers with Antibodies in LFAs

In recent years, nucleic acid-based diagnostic tests have attracted considerable attention. Oligonucleotides can not only be employed as probes for the detection of complementary DNA or RNA targets, but they can also use affinity probes or bio-recognition elements for different targets such as small molecules, metal ions, lipids, peptides, proteins, sugar moieties and whole cells [20]. Nucleic acid-based scaffolds with the ability to bind to a variety of target molecules are referred to as aptamers. Aptamers are short single-stranded DNA or RNA (often DNA because of greater stability than RNA, which makes the aptamer selection process easier) with molecular weight ranging between approximately 10 and 30 kDa. They are a promising group of bio-recognition elements. Aptamers are synthesized by an in vitro selection process named SELEX (systematic evolution of ligands by exponential enrichment) [21]. Aptamers can bind to target analytes by virtue of their unique three-dimensional shape via binding forces such as Van der Waals forces, hydrogen bonding, stacking of aromatic rings, salt bridges, other electrostatic interactions, and shape complementarity [22]. Due to their chemical features, aptamers are known as “chemical antibodies”, indicating their functional similarity to protein antibodies [22]. Aptamers can successfully compete with antibodies because of high affinity and specificity towards their target. Compared to the conventional antibodies, aptamers possess several advantages such as easy artificial synthesis, lack of immunogenicity, cell-free evolution, higher stability, and adaptability to various targets [22,23]. In recent years, different types of aptamer-based sensors and assays have been developed to detect a variety of analytes [24,25,26,27]. In these assays, target-binding signals are detected using optical or electrochemical systems. Aptamer-based sensors/assays show several advantages such as high stability and reusability, more flexibility in designing different sensing platforms and multiplexed assays, and more access to biological compartments in disease diagnostic applications due to the small size of aptamers compared to antibodies [23]. Due to the inherent advantages of aptamers, they can be a good alternative to antibodies in designing LFAs. Aptamer-based LFAs open new horizons for designing novel diagnostic platforms. For instance, aptamers can hybridize with complementary DNAs and deconstruct hybridization once they meet a target molecule. Moreover, in some detection systems, aptamers can regenerate by heating or other methods. Such advantages make aptamers appealing for developing low-cost, reusable, and robust analytical tools. 

Similar to antibody-based LFA, there are two formats of aptamer-based LFA including sandwich and competitive formats. Moreover, there are some other aptamer-based LFAs with distinctive nucleic acid features.

In a typical example of sandwich format, two aptamers are bound to analytes at two different sites (Figure 2a). The primary aptamer is thiolated and conjugated to AuNPs through the S–Au binding affinity. The AuNPs-labeled aptamer is used as a recognition element and dried on the conjugate pad. The biotinylated secondary aptamer is served as a capture aptamer and sprayed on the test line via biotin-streptavidin binding. When the sample solution containing analyte is dropped on the sample pad and flows into the conjugate pad, the analyte binds to the AuNP-labeled aptamer. Then, the complex migrates along the strip towards the nitrocellulose membrane, where the analyte part of the complex is captured by the secondary aptamer in the test line. The excess AuNP-primary aptamer that has passed the test line is captured in the control line through hybridization between the immobilized DNA probes in the control zone and the primary aptamer. Therefore, two red bands in the test and control zones indicates a positive result, while only one red band is observed in the control zone for negative samples [2,23,28]. Different types of sandwich LFA using aptamer as recognition elements have been developed, some of which will be discussed in the following sections.

In a common competitive format for the detection of small molecules such as mycotoxins with only one aptamer, aptamers with a poly A tail are conjugated with AuNPs in a similar way to the method described in the sandwich format (Figure 2b). A biotin-modified complementary strand (DNA probe 1) is immobilized via streptavidin binding onto the test zone. On the other hand, a biotin-modified poly DNA probe 2 is immobilized onto the control line through streptavidin binding. In this platform, the principle of detection is based on a competitive reaction between DNA probe 1 and the target molecule for binding to aptamers. In the presence of analyte in the sample solution, analyte is bound to the AuNP-labeled aptamer, decreasing the concentration of AuNP–aptamer that can hybridize with DNA probe 1 on the test line and resulting in a weaker intensity of colorimetric signal [23,29]. In fact, in this format, the concentration of analyte in the sample is inversely proportional to the intensity of the color signal. The unbounded AuNP–apatmer hybridizes with DNA probe 2 on the control line. Compared to antibodies, aptamers represent increased flexibility in competitive LFAs.

## 4. Clinical Applications of Aptamer-Based LFAs

Generally, clinical diagnostic tests are performed in medical diagnostic laboratories and the results will be ready after several hours or even days. In many cases, a timely decision can strongly affect the clinical outcome, and so the potential benefits of LFA in the clinical field are obvious. This technique can be used in all clinical aspects including screening, diagnosis, prognosis, monitoring, and surveillance. Rapid clinical evaluation has a major impact on disease management, reducing workload, increasing workflow, improving medical care, and reducing treatment costs. The use of LFA allows the patient to receive a quick diagnosis and treatment during the same consultation, reducing the number of visits to the doctor and avoiding problems related to delaying the start of treatment [30,31]. Furthermore, the long-term benefits of this method should not be underestimated. For instance, LFA is able to differentiate between viral and bacterial infections and helps in the decision to prescribe antibiotics where needed, thus limiting the misuse of these drugs, which can lead to bacterial resistance. The great success of LFA is due to its great impact on human health and also because the first applications of this method had clinical purposes (i.e., the pregnancy test). In addition, it is worth noting that even with the relative complexity of biological fluids, their number is very limited. For instance, the most commonly used biological matrices include saliva, venous or capillary blood, urine, nasopharyngeal swabs, and stools. Therefore, after proper sample pretreatment, the analysis should be very easy. Depending on the sample type, the treatment can be very different and includes plasma separation from fingerstick or whole blood, filtration, cellular lysis for bacteria or viral intracellular antigens, changes in pH of urine, or breakup of mucins in saliva or respiratory samples [32]. In most cases, these treatments are performed using special sample pads in the strip and therefore do not require additional steps [31,33]. Remarkably, the bio-recognition elements used in LFAs, including antibodies and, more recently, aptamers, are able to perform well in biological fluids.

Over the past ten years, 69% of LFA strips have been used to detect infectious diseases, 28% to detect endogenous markers and biomarkers, and 3% to detect drugs [32]. As can be seen, the most common use of these tests is to diagnose infectious diseases, which is due to the importance of rapid and early detection of these diseases and prevent their spread. The importance of LFAs led them to be included in the list of essential in vitro diagnostic (EDL) tests by the World Health Organization (WHO) in 2018 [32]. Due to the obvious importance of LFAs in diagnostic applications, some clinical applications of this method with emphasis on aptamers as a new generation of bio-recognition elements will be discussed below. 

### 4.1. Detection of Biomarkers

Biomarkers or biological markers are measurable indicators that determine severity or presence of some particular disease state, physiological state or pharmacologic responses to a specific treatment in the body [34,35]. They are very important diagnostic tools that can be measured in biological fluids such as blood, sweat, urine, or soft tissues to evaluate normal biological processes. Recently, rapid and low-cost detection of biomarkers has received increasing attention. In this regards, LFA technology with unique properties such as ease of operation and fast turnaround time are increasingly being used as POC diagnostic tests for detecting a wide range of biomarkers. Considering that in antibody-based LFAs, specificity and sensitivity can be affected by other chemicals with similar structures, leading to false positive results [1], and because of the other disadvantages of the antibodies mentioned earlier, in recent years, the application of POC tests in the detection of biomarkers has shifted to the use of aptamers as recognition elements in the design of sensing platforms, including LFA technology. Aptamers are generally applied in conjunction with AuNPs for simple sensitive colorimetric detection in POC devices such as LFAs [36]. Aptamer-modified AuNPs have been used for the detection of different analytes in biological fluids. For example, Dalirirad et al. (2020) incorporated conjugated AuNP–aptamers in LFA for easy and sensitive detection of dopamine, a critical biomarker for determining stress levels, in urine [37]. The principle of detection was based on the “duplex dissociation mechanism” (Figure 3a). In this method, a capture aptamer (DNA3) that was conjugated with the AuNPs through a 20T linker (DNA1) could hybridize with a part of a free aptamer (DNA2) in solution in order to act as a sensor probe. In the absence of dopamine, two aptamer strands were hybridized, while in presence of dopamine, DNA2 undergoes conformational changes resulting in dissociation from DNA3 and binding to the target molecule. This step occurred in a solution phase. On the nitrocellulose membrane, test and control lines were coated with streptavidin, which bound to the biotinylated complementary oligonucleotide strands. Test and control lines were functionalized with cDNA3 and cDNA1 that were complementary to the capture part of the duplex and to the linker, respectively. In the presence of dopamine in sample solution and dissociation of DNA2 from DNA3, the capture probe was hybridized to the cDNA3 in the test line. In the absence of dopamine, DNA3 remained as a part of the duplex and resulted in no hybridization with the immobilized cDNA3 on the test line. On the control line, a red band was generated by the hybridization between thiolated 20T DNA1 and biotinylated 20A cDNA1. The visual limit of detection (LOD) was obtained at approximately 50 ng mL^−1^ (normal concentration of dopamine in urine is in the range of 52–480 ng mL^−1^ or 0.3–3.13 μM). Moreover, the assay showed high selectivity. The stability of aptamers and AuNPs with no degradation or aggregation was only 35 days while the intensity of the test line started to decrease after 25 days of storage. However, due to the high stability of aptamers, this period can be increased by changing the storage conditions.

In another study, AuNPs were used as a label in an adsorption–desorption mechanism to detect human epidermal growth factor receptor 2 (HER2) in serum samples [38]. The overexpression of this factor indicates breast cancer, so that the concentration of HER2 in the blood of breast cancer patients increases to 220 ppm–1.1 nM compared to 30 pM–220 pM in normal persons. Moreover, due to the relationship of overexpression of this factor with other cancers such as gastric, lung, ovarian and oral, HER2 is used as an important biomarker in the early diagnosis of cancer [39]. For simple and rapid detection of HER2, Ranganthan et al. (2020) developed an aptamer-based LFA with high sensitivity (Figure 3b) [38]. In this work, the LFA strip was made with a coating of streptavidin and pullulan mixture at the test line and a cationic charged polymer named poly (diallyldimethylammonium chloride) (PDDA) at the control line. Aptamer–AuNP complexes were prepared by adsorption of biotin-modified HER2 aptamers. In the presence of target analyte in the sample solution, HER2 specifically bonded to its aptamer resulted in release of the AuNPs surface. Then this solution was applied on the LFA strip, which led to the absence of a red signal on the test zone as the aptamer–HER2 complex was bound to the streptavidin on the membrane. The control zone turned red due to non-specific adsorption of anionic AuNPs by the cationic PDDA. In the absence of analyte, the aptamer–AuNP conjugate remained as the intact complex leading to generation of a red signal on the test zone due to aptamer–AuNP binding to the streptavidin. The control zone turned red as before. The assay LOD was obtained at 20 nM. The detection time was estimated to be approximately 30 min. The stability of the assay was not evaluated. 

**Figure 3 pharmaceuticals-15-00090-f003:**
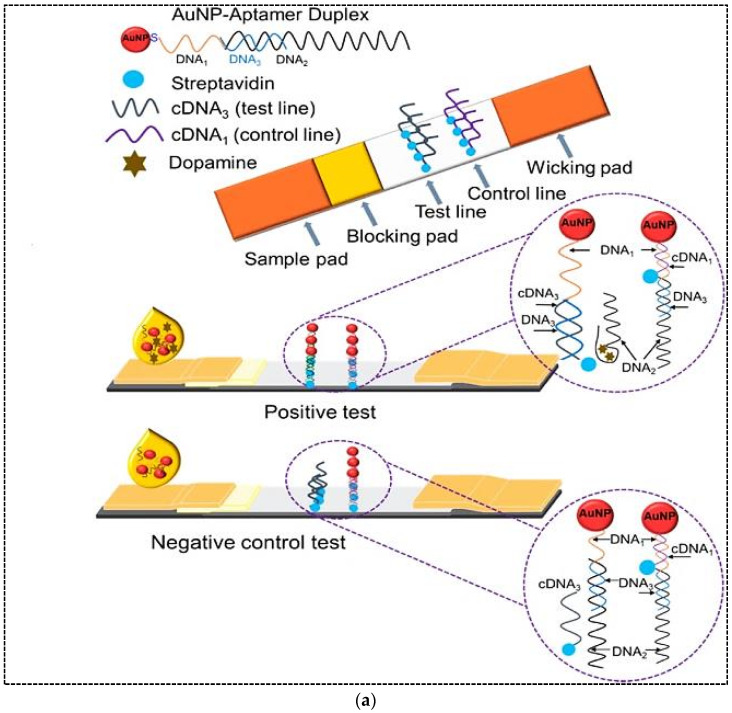
(**a**) A schematic representation of the duplex dissociation mechanism for the detection of dopamine. (**b**) Adsorption–desorption colorimetric LFA for HER2. (**c**) An aptamer-based competitive LFA using gold nanozyme as a label for the detection of CA125. (**d**) An aptamer-based hook-effect-recognizable three-line lateral flow biosensor for rapid detection of thrombin. Reprinted with permission from [37,38,40,41], respectively.

Several strategies can be used for increasing the sensitivity of LFAs in order to detect very low concentrations of analyte. Changing the label and replacing AuNPs with fluorescent labels or using enzymatic reactions to generate a colorimetric signal is one of the most common and popular methods to increase sensitivity. In this regard, enzyme mimetic nanomaterials or nanozymes with intrinsic enzymatic activity are a good option due to their unique properties compared to natural enzymes. Nanozymes overcome the disadvantages of natural enzymes and present an affordable alternative to them. They show high catalytic activity, stability and multifunctionality. Tripathi et al. (2020) used gold nanozyme with peroxidase-like activity in an aptamer-based competitive LFA for rapid detection of CA125 (ovarian cancer serum biomarker) in serum [40]. In this study, a CA125-specific aptamer was immobilized on a nitrocellulose membrane and used as a capture probe (Figure 3c). A certain concentration of CA125 conjugated with AuNPs competed with varying concentrations of unlabeled CA125 in the sample for binding to the immobilized capture probe. Captured AuNPs on the test line were applied as a peroxidase mimetic to catalyze the oxidation of 3,3′-diaminobenzidine (DAB)/H_2_O_2_ substrate. The color intensity at the test line was inversely related to the concentration of CA125 in the sample. The assay showed high specificity and sensitivity with a LOD of 5.21 U ml^−1^ and an assay time of 20 min. The authors estimated the cost of manufacturing each strip as less than 1 USD. Despite all the advantages of the designed method, the authors did not mention the stability of the strips during storage.

The sandwich format of LFA is the most widely used format in both antibodies-based and aptamers-based assays for detection of a variety of analytes. However, in this format, the detection range is often limited by a phenomenon called the hook effect. The hook effect is a false negative phenomenon resulting from an improper ratio of antigen and antibody. The effective factor in the high-dose hook effect is that the excess amount of unlabeled analytes has already occupied the reaction sites on the test line. In this case, as the analyte concentration increases, the test line signal does not increase or even decreases [41]. Therefore, such a sandwich LFA cannot quantify analytes in a relatively high concentration range. To address this issue, Gao et al. (2019) developed a sandwich format of aptamer-based LFA that was able to eliminate the hook effect and make the detection over a wide range [41]. In this method, a three-line LFA was designed by adding a thrombin line (TB line) between test and control lines (Figure 3d). The TB line could help to distinguish between high and low doses of analyte when the hook effect occurred. A biotinylated aptamer with a poly A tail (T-DNA) and a biotinylated poly T oligonucleotide probe (C-DNA) were immobilized on the test and control lines, respectively, through binding to immobilized streptavidin. AuNPs were used as a label and conjugated with thiolated aptamer with poly A tail (AuNP–Apt1). In the presence of ultrahigh concentration of thrombin, there would be a large amount of unreacted thrombin in the sample after the binding of Au–Apt1 conjugates to thrombin. In this case, the migration rate of unreacted thrombin would be faster on the strip due to its smaller size compared to the thrombin–AuNPs–Apt1 complex. Therefore, unreacted thrombin molecules occupied the reactive site on the test line in advance, and the sandwich detection of thrombin was inhibited, which resulted in a weak colored signal on the test line. When the mixture continued to migrate through the target line, most of the AuNP–Apt1 conjugates were bonded to excess thrombin from the sample solution. Therefore, it inhibited the reaction with thrombin on the target line, resulting in a weak signal. AuNP–Apt1 conjugates and thrombin were captured on the test line due to the recognition reactions between the aptamer and thrombin. But AuNPs–Apt1 was captured by the control line. Consequently, the response on the control line was unaffected. A large number of AuNP–Apt1 conjugates were captured by the control line and a stronger signal was generated on the control line. A correspondence table between the TB concentration in the sample and the intensity of the three lines on the strip is shown in Figure 3d. Indeed, the thrombin concentration in the sample was quantitatively related to the signal formation on the three lines of the strip. The LOD of the new assay was 0.85 nM with a wide linear range of 1 nM–100 µM. The assay time was estimated to be 10 min. Moreover, the LFA strip showed high accuracy and reproducibility. However, the stability of strips during storage was short and the signal intensity decreased after 3 weeks.

Representative examples of recent developed aptamer-based LFAs for the detection of biomarkers are listed in Table 1.

### 4.2. Detection of Microbial Analytes

The rapid and sensitive detection of pathogenic bacteria viruses is crucial for public health, medical diagnostics, environmental monitoring, and food safety fields in order to avoid outbreaks. Conventional detection methods relying on culturing and biochemical analysis are time consuming and take 2 or 3 days or longer [46]. For instance, it is clear that virus cultivation is tedious work, which requires well-trained personnel and specific methodologies [47]. Currently, detection methods for pathogenic bacteria and viruses include polymerase chain reaction (PCR) and enzyme-linked immunosorbent assay (ELISA). Despite many advantages of these methods including high accuracy and decent selectivity, they suffer from drawbacks of being time consuming and labor intensive [47,48]. Moreover, they need expensive instruments and they are not suitable for use in deprived areas [47]. To overcome the limitations associated with conventional diagnostic methods, alternative robust approaches have been developed for bacteria and virus detection with high sensitivity and accuracy and low detection time. Among all types of biosensors and other diagnostic methods developed to detect bacteria and viruses, similar to other analytes, LFA is particularly popular. Most of these LFA methods have been developed to identify pathogenic bacteria and viruses that have caused many health and economic problems worldwide. For example, Ren et al. (2021) developed an aptamer-based LFA for sensitive detection of *Escherichia coli* O157:H7 (*E. coli* O157:H7), which is one of the most widespread foodborne pathogens [49]. In order to increase the assay sensitivity, they combined aptamer-exonuclease III (Exo III)-assisted amplification with AuNPs-based LFA. The Exo III can hydrolyze the 3′ recessed end or blunt end of dsDNA, while it has limited activity for ssDNA or dsDNA with 3′-terminal producing. First, the short target ssDNA was hybridized to the anti *E. coli* O157:H7 aptamer. In the presence of *E. coli* O157:H7, the target bacteria was bound to its specific aptamer and ssDNA was released and hybridized with the probes of the designed hairpin (HP). Exo III digested the 3′ double-stranded blunt end of the complex and released the enzyme product. Because the remaining sequence of the HP of the enzyme product was the same as the target ssDNA sequence, the target ssDNA could be amplified. Eventually, the amplified target ssDNA was applied to the LFA system to achieve visual detection of *E. coli* O157:H7. Indeed, one end of the target ssDNA was hybridized with the AuNP probes and the other end was hybridized with the capture DNA probe on the test line to obtain a colored signal. The unbound AuNP probes migrated towards the control line and were hybridized with their complementary probes immobilized on the control line. The quantitative ability of the designed assay for pure culture was 7.6 × 10^1^ cfu mL^−1^, while the LOD in milk was 8.35 × 10^2^ cfu mL^−1^. The detection time was 4 h, which was comparable to conventional culture methods and molecular assays.

In another study, a LFA based on aptamer-gated silica nanoparticles and release of signal molecules was developed for highly sensitive detection of *Listeria monocytogenes* [50]. Aptamer-gated assays are based on the opening mechanism established on the molecular conformational changes of aptamers to release a reporter molecule entrapped in pores of mostly silica nanoparticles. As shown in Figure 4b, the mesoporous silica nanoparticles were loaded with 3,3′,5,5′-tetramethylbenzidine (TMB). Then, the amine functionalized aptamers were immobilized on the particles through the epoxy reaction. Capping of silica nanoparticles with aptamer gate sequences blocked the pores of the particles and prevented the release of TMB. On the other hand, the horseradish peroxidase (HRP) was immobilized on the test zone of the nitrocellulose membrane by physical adsorption. TMB-loaded aptamers-gated silica nanoparticles were immobilized on the conjugate pad. After addition of sample containing bacteria cells in the sample pad, target bacteria were bound to aptamer-gated silica nanoparticles in the conjugate pad causing molecular conformation changes on the aptamer structure, blocking the leakage of TMB from nanopores of mesoporous silica. The conformational change of aptamers resulted in specific release of TMB from mesoporous silica. TMB migrated to the test zone and interacted with the immobilized HRP to produce a blue precipitate resulting from an oxidation reaction with peroxide. The LOD of the method was 53 cells mL^−1^ with a detection time of only 5 min. The label-free strategy was employed to detect *L. monocytogenes* in minced chicken samples.

Avian influenza viruses are highly pathogenic viruses that can infect almost every animal in the world including humans. They have pandemic potential because of mutations originating from their instability of their RNA. In order to prevent the economic losses caused by the outbreak of bird flu as well as human pandemics, rapid and accurate detection of this virus is of great importance. The LFA technique is considered to be appropriate for the on-site detection of influenza viruses. Although the sandwich format of LFA is highly stable, sensitive, and specific compared to the competitive format, it is used only for large analytes. In order to design a sandwich LFA, Kim et al. (2019) developed a cognate pair of aptamers with the ability to bind to the different sites of influenza viruses using immobilization-free graphene oxide-SELEX [51]. The cognate pair of aptamers was applied on a sandwich LFA strip for the detection of whole virus particles. The test line was coated with biotin-modified primary aptamer through streptavidin/biotin binding (Figure 4c). The biotin modified poly A sequences were immobilized on the control line. For the signal generation, AuNPs were conjugated with thiol and poly T modified at the 5′ end of the secondary aptamer. In the presence of virus particles, a sandwich complex of primary aptamer–virus–secondary aptamer/AuNPs conjugate was formed on the test line. The unbound AuNP–poly T modified secondary aptamer conjugates moved towards the control line and hybridized to A sequences immobilized on the control line. Upon optimized conditions, the LOD was obtained at 6 × 10^5^ EID_50_ mL^−1^ (EID_50_/mL: 50% egg infective dose) in buffer and 1.2 × 10^6^ EID_50_ mL^−1^ in the duck’s feces, respectively, by naked eye, while the quantitative LOD was 1.2 × 10^5^ EID_50_ mL^−1^ in buffer and 2.09 × 10^5^ EID_50_ mL^−1^ in the duck’s feces. Due to application of a sandwich LFA, the specificity towards the target virus H5N2 was significantly increased. The assay time and stability of the strips during storage were not reported by the authors.

In another study, for the detection of multiple strains of influenza virus, a dual recognition element LFA, which paired an antibody with an aptamer, was developed [52]. The combination of two kinds of recognition elements was used to overcome the individual limitations of antibodies’ cross-reactivity and aptamers’ slow binding kinetics. In this study, both biotin-modified specific aptamers and AuNPs conjugated with monoclonal antibody (mAb) detected virus particles and formed a complex of aptamer–virus–mAb/AuNPs. The complex was captured by streptavidin at the test line through the biotinylated aptamer and produced a colored signal via AuNPs. The unbound AuNP–mAbs were captured by secondary antibodies printed at the control line. In the absence of target, the complex was not formed and therefore no signal of AuNPs was observed on the test line. This method provided fast capture kinetics on both test and control lines. The LOD was found to be 2 × 10^6^ virus particles. The dual recognition LFA was able to detect virus strains of other subtypes with high specificity. The detection time and storage stability were not reported. 

Another microbial analyte that is affecting the world these days and whose diagnosis is extremely important is the SARS-CoV-2 virus. In order to manage the spread of the virus and reduce community transmission, rapid diagnostic tests, vaccine, specific medicines, and intelligent tracking strategies are absolutely essential. In this regard, LFA tests play an important role in the rapid diagnosis and prevention of disease spread. Since the COVID-19 outbreak in 2019, researchers around the world have made great efforts to design and develop rapid diagnostic tests. The design and development of lateral flow tests has been one of the most successful of these efforts. Hundreds of LFA-based diagnostic prototypes have emerged, some of which have been developed into commercial diagnostic kits for the rapid detection of COVID-19. In this regard, different types of LFA have been developed that target SARS-CoV-2 specific biomarkers such as viral RNA, antibodies, antigens, and whole virus. Apart from LFAs developed for viral nucleic acid detection that rely on the integration of nucleic acid amplification technologies, such as RT-PCR, isothermal amplification, and clustered regularly interspaced short palindromic repeats (CRISPR) with LFA [53], other developed LFAs are based on antigen–antibody interactions. In fact, to date, no aptamer-based LFA has been developed to diagnose SARS-CoV-2 infection. Therefore, conducting studies in this field in order to possibly improve LFA-based diagnoses for rapid detection of SARS-CoV-2 is of particular importance. Developing LFA-based technologies for targeting the SARS-CoV-2 mutants is crucial for preventing further spread. In this regard, the performance of aptamers can be compared with that of antibodies in detecting mutants.

Representative examples of recent developed aptamer-based LFAs for the detection of microbial analytes are listed in Table 2.

### 4.3. Detection of Hormones

Hormones are chemical messengers that are secreted directly into the blood, which carries them to different organs and tissues to regulate their functions. In vertebrates, hormones are responsible for the regulation of many physiological processes and behavioral activities such as digestion, metabolism, respiration, sensory perception, lactation, growth and development, and reproduction. Changes in hormone levels (hormone imbalance) can lead to irreversible side effects. Detection and evaluation of hormone imbalance can help treat the disease and reduce its complications. Hormone testing can be conducted using blood, urine or saliva samples. Traditional detection methods such as ELISA, chemiluminescent immunoassay (CLIA), and electro-chemiluminescence immunoassay (ECLIA) require specialized equipment and professional personnel and are time-consuming, making them unsuitable for use in POC testing. After the first use of LFA in 1976 to detect human chorionic gonadotropin (hCG) in urine, to date, this sensing platform has been developed and applied to detect many other hormones. Dalirirad and Steckl (2019) developed an aptamer-based LFA for POC detection of cortisol in sweat [56]. Cortisol in sweat has been recognized as a main biomarker to monitor physiological stress. In this study, AuNPs were functionalized with cortisol specific aptamers (Figure 5a). The minimum number of aptamers required to cover AuNPs and prevent aggregation against salt (NaCl) was determined. Cysteamine molecules were immobilized on the test zone of a nitrocellulose membrane. In the presence of cortisol molecules in sweat samples, they bound to aptamers resulting in aptamer desorption from the AuNP surface. Then, free AuNPs were captured by immobilized cysteamine through affinity of AuNPs to the thiol group of cysteamine. A colored signal was generated on the test line within minutes. In the absence of cortisol, shielded AuNPs in solution migrated through the membrane and did not interact with cysteamine. The assay exhibited a visual LOD of 1 ng mL^−1^, which covered the normal range of free cortisol in sweat (8–140 ng mL^−1^). The shelf life of the strips and the AuNP solution were evaluated after approximately 10 days and 30 days, respectively. However, longer-term testing is required for commercial test development.

In another study developed by the same author, cortisol was detected in saliva with a higher sensitivity compared to the previous work [57]. For the fabrication of sensor probes in LFA, AuNPs were conjugated with a duplex aptamer including thiol- and ploy T-modified capture probe and cortisol aptamer through Au–S bonds. One end of the capture probe was complementary with a part of the aptamer as well as the biotin-modified oligonucleotide probe immobilized on the test line, while the other end (poly T tail) was complementary with a biotin-modified poly A probe immobilized on the control line (Figure 5b). The addition of saliva sample containing cortisol induced conformational changes of the cortisol–aptamer causing aptamer dissociation from the capture probe. Then, the single stranded capture probe was hybridized with an oligonucleotide probe immobilized on the test line to produce colored signal. The unbound AuNP–capture probe migrated along the nitrocellulose membrane and hybridized with poly A probe immobilized on the control line. The LOD was 0.37 ng mL^−1^ with a linear range of 0.5–15 ng mL^−1^. The shelf life of the strip was estimated to be two months, which is relatively acceptable.

Endocrine disrupting compounds (EDCs) are natural and synthetic exogenous compounds with estrogenic activity that can mimic the actions of the native hormones, resulting in serious disruption of the reproductive system. Pharmaceutical and industrial growth is the main reason for the spread of these compounds in the environment. In particular, progesterone (P4) is a multifunctional steroid hormone with a variety of medicinal uses, and can enter the environment and contaminate surface water. For the detection of P4 in environmental samples, a competitive LFA was developed by Alnajrani and Alsager (2019) [58]. In this assay, AuNPs were conjugated with a 60-mer P4 aptamer and further hybridized with a biotinylated 8-mer complementary sequence to form AuNP-duplexed aptamers. In the absence of P4, the biotinylated sequence of the AuNP-duplexed aptamer interacted with the immobilized streptavidin on the test line and produced a red signal. Under addition of sample containing P4, the aptamer underwent a conformational change that resulted in dissociation of the 8-mer biotinylated sequences and reduction of signal intensity on the test corresponding to the target concentration. The LOD of the assay was found to be 5 nM in buffer and tap water. Moreover, the designed method showed high specificity. However, storage stability and complex matrices were not evaluated. 

Unfortunately, despite the relatively long time since the emergence and use of aptamers, their clinical application in LFA strips for the detection of hormones has not been adequately investigated.

Representative examples of recently developed aptamer-based LFAs for the detection of hormones are listed in Table 3.

### 4.4. Detection of Antibiotics

Antibiotics are antimicrobial substances produced by microorganisms, animals and plants. They possess antimicrobial activity against pathogenic microbial species and can be widely used in human and veterinary medicine to treat many infectious diseases [59]. Large amounts of antibiotics are used annually in human disease treatment, livestock industry and aquaculture worldwide, whereas their misuse results in sustainable side effects in human health and the environment. The presence of antibiotics in the environment leads to an increase in the number of multi-resistant bacteria, with subsequent serious health risks for human and animals [60]. Therefore, sensitive analytical methods are required for the determination of antibiotics in food and the environment. The most common methods for the detection of antibiotics are chromatographic techniques, electrophoresis and ELISA [61]. Due to the challenges associated with these methods, as previously discussed, the need for rapid, accurate, sensitive and on-site methods is essential. In recent years, antibody-based LFA strips have been commercially developed for rapid and quantitative detection of antibiotics. However, aptamer-based LFAs for antibiotic detection are still being studied and researched. Ou and coworkers (2019) developed a strip assay based on magnetic pretreatment and AuNP probes for rapid and highly sensitive detection of kanamycin [62]. In this study, the concentration of kanamycin in the sample was converted to the concentration of kanamycin-displaced cDNA from magnetic microspheres (MMS). The assay was divided into two steps (Figure 6a). First, the aptamer–cDNA duplex was formed and conjugated with MMS through biotin–streptavidin binding. Upon addition of sample containing kanamycin to this mixture, kanamycin was bound specifically to the aptamer and released cDNA from the duplex. The released cDNA was positively proportional to kanamycin concentration; therefore, the supernatant after magnetic separation was applied on the strip. On the strip, AuNP–DNA1 conjugate was immobilized on the conjugate pad, while biotin-modified capture DNA1 (complementary to 5′-end of cDNA) and biotin-modified capture DNA2 (complementary to 3′-end of cDNA) were immobilized on the test and control lines, respectively, through streptavidin–biotin binding. The 5′-end of cDNA was complementary to DNA1 and the 3′-end of cDNA was complementary to capture DNA1. Therefore, upon addition of cDNA onto the strip, cDNA in the detection solution could mediate the hybridization and formation of sandwich structure of capture DNA1/cDNA/DNA1–AuNP conjugates on the T line. Since capture DNA2 was complementary to DNA1, the excess DNA1–AuNP conjugate was hybridized to DNA2 on the C line. The visual and quantitative LOD of the assay were 50 nM and 4.96 nM, respectively, with a linear range of 5–500 nM for quantitative analysis. The LFA strips were applied for the detection of kanamycin in food samples including milk and milk products and honey. The assay time was only 20 min. The stability of the biosensor during storage was estimated to be 3 months.

In another study, a competitive LFA was developed for oxytetracycline (OTC) detection in milk [63]. To develop strips, OTC-carrier protein conjugate and streptavidin-complementary DNA probe conjugate were immobilized on the test and control zones, respectively. AuNPs were conjugated with OTC specific aptamers. Upon addition of sample containing OTC, the free OTC was bound with the AuNPs-aptamer conjugate and formed AuNP–aptamer–OTC complexes, which moved across the nitrocellulose membrane. Depending on the OTC concentration in the sample, the unbound AuNPs could bind with the immobilized OTC at the test line. Therefore, a red band was formed in the test line, where its color intensity was inversely proportional to the OTC concentration in the sample. The red band at the control line was generated due to complementary binding of AuNP–aptamers with the DNA probe. The assay was able to detect up to 5 ng mL^−1^ of OTC in spiked milk within 10 min. Stability of the strips during storage was not evaluated. 

In another study, a competitive LFA based on the cross-reaction of an aptamer for ampicillin with C-reactive protein (CRP) and vice versa on a CRP aptamer also binding with ampicillin was developed [64]. Indeed, both aptamers against ampicillin and CRP showed partial sequence overlap, showing that both aptamers could bind to the same targets. Therefore, this cross-recognition was used for the development of a label-free LFA for ampicillin detection. In this platform, AuNPs were dually conjugated with aptamer and mouse Fc fragment (mFc). The biotin-modified CRP was used as a competitive agent. Streptavidin and α-mouse antibody were immobilized on the test and control lines, respectively (Figure 6c). In the absence of ampicillin, aptamers on the AuNP–aptamer–mFc conjugates bound to the CRP, resulting in a strong signal on the test line. Unbound AuNP–aptamer–mFc probes flowed further through the strip and were bound to the α-mouse antibodies on the control line through interaction between the mFc and α-mouse antibodies, resulting in a reliable control line. In the presence of ampicillin in the sample, it competed with the CRP for binding to the aptamer, resulting in a decreased signal on the test line. The LOD was found to be 185 mg L^−1^. The strips were used for the detection of ampicillin in milk samples. The shelf life of strips was not evaluated. 

Representative examples of recently developed aptamer-based LFAs for the detection of antibiotics are listed in Table 4.

## 5. Evaluation of Sensitivity and Specificity of Aptamer-Based LFAs in Comparison with Antibody-Based LFAs

Sensitivity and specificity are two important factors in the design of diagnostic tests including LFAs. Numerous studies have been performed to increase the sensitivity and specificity of LFAs as the popular rapid diagnostic tests. Given the new approach of researchers to the use of aptamers as emerging recognition elements, the question arises in the minds of many researchers of whether the use of aptamers can improve the sensitivity of these tests? According to the results of many studies, it can be concluded that aptamers are examples of “programmable” nucleic acid-based binding reagents that have a range of sensitivities from mid-picomolar to high-micromolar, which argues for a constant need for case-by-case optimization even though—or especially because—they introduce new reaction sets to the LFAs. Since the better analyte capture can increase the assay sensitivity, it can be assumed that the use of aptamers in the detection of some analytes, especially small molecules that are challenging to detect using antibodies, can provide better sensitivity. In the case of other analytes, there may not be a significant difference in sensitivity between aptamer-based and antibody-based LFAs and method optimization and application of other strategies (e.g., changing the label, the type of nitrocellulose membrane, or using an enzymatic reaction to amplify the detection signal) that can increase the sensitivity of the method regardless of the type of recognition element. Moreover, in aptamer-based LFAs, high sensitivity can be received by performing signal amplification using polymerases and/or nucleases, longer (up to 200 base) “multivalent” aptamers. Therefore, the selecting type of biorecognition needs to be carefully considered.

Regarding specificity of antibody-based and aptamer-based LFAs, it is really difficult to produce high-affinity antibodies for the detection of small molecules and toxins, thus aptamer technology can fill this gap easily by designing and developing high-affinity aptamers against these analytes and adapting them on to the LFA format. Overall, the aptamer-based LFAs show a bright future for analytical applications due to their broad potential and ability to be adjusted on demand.

## 6. Current and Future Challenges of LFAs

LFAs have been introduced and developed over the last half century. During this time, this method became very popular among other diagnostic methods. POC diagnosis using LFAs has attracted the attention of the research community, especially in the development of platforms for the detection of various analytes in body fluids such as saliva, urine and sweat. Despite all the advantages, this method, with half a century of history, still faces challenges. Limited sensitivity in some applications with low concentration of analyte is one of the main challenges. The use of alternative materials to better capture analyte and/or amplify the signal by changing the label can address this issue. Although various studies have been conducted in this field, so far only a few of these studies have reached the commercialization stage. Another challenge is the need for sample pretreatment in some applications, such as the use of strips to detect analytes in the blood. In fact, LFA strips are not ideal for complex matrix or solid samples. In these cases, sample extraction, filtration or even dilution is necessary before testing. LFA tests cannot yet be used completely independently and with sufficient reliability. Therefore, confirmatory analysis is still needed in cases where the test is positive. Multiplex detection is still not common due to concerns about cross reactivity. The nature of these tests is qualitative or semi-quantitative, and so far many efforts have been made to quantify the results, some of which have led to the development of portable readers for semi-quantitative detection of some analytes. However, technological improvement of the LFAs in the future can increase the cost of analysis and make them unsuitable for home use or deprived areas. One of the concerns that may arise in the future regarding these user-friendly tests is the issue of self-diagnosis and self-medication and failure to visit medical centers at the appropriate time, which may harm people’s health. Other future challenges of LFAs include bad reputation in case of misuse, possible cross reactivity and possible matrix interference, which results in misdiagnosis.

## 7. Conclusions

The LFA technique has made a huge change in various diagnostic fields due to ease of operation, quick response, accuracy, and sometimes quantitative results. Currently, this method can be used to detect a variety of analytes in medicine, agriculture, food and environmental safety, etc., and its test strips are produced by many companies around the world. Although the principles of this method have remained unchanged for decades, continuous improvements have been made that lead to increased sensitivity, reproducibility, stability, and simultaneous identification of multiple analytes. One of the main limitations of this technique is high inter-batch variability due to the use of antibodies. This limitation can be easily overcome by using other synthetic recognition elements such as aptamers. Compared to antibodies, aptamers are selected through an in vitro process (SELEX), thus avoiding the use and killing of animal hosts [66]. This process allows the selection of aptamers against non-immunogenic molecules. Although the SELEX process is costly, after screening aptamers, their synthesis cost is significantly lower than that of antibodies. Moreover, aptamers have higher reproducibility and chemical stability, which prevents batch-to-batch variations and eliminates the need for storage at low temperatures [67,68]. Due to the inherent nature of nucleic acids, aptamers can also regain their structure after denaturation, allowing them to flexibly adapt to different assay formats that are not available with antibodies [69]. All of these features make aptamers suitable candidates for replacement with antibodies in LFAs. This review discussed current trends in clinical diagnostic rapid tests relying on aptamer-based LFAs. As can be concluded, despite the salient advantages of aptamers, in the field of identification of a variety of analytes, especially in clinical samples, using aptamer-based LFAs, few studies have been performed. For example, the number of published articles on the detection of hormones and antibiotics is very limited. Among them, published articles on antibiotics have made the diagnosis in food matrices, not clinical samples. Generally, the application of aptamers in LFAs is considered as proof-of-concept for new on-site biosensors, and only a few aptamers have been successfully used in LFAs. The type of nitrocellulose membrane used, its pore size, and the variation in label type have not yet been investigated in aptamer-based LFAs. Immobilization of capture and control molecules on the membrane in the test and control zones plays an important role in membrane performance. However, so far, the affinity reaction between streptavidin and biotin is the most extensively used technique for immobilization of aptamers on the nitrocellulose membrane. Furthermore, currently, AuNPs are the labels that have been widely studied in aptamer-based LFAs, while the application of other labels can affect the assay sensitivity. In some studies, several strategies including the use of silica nanoparticles, magnetic separation and enzyme amplification has been employed to enhance the assay sensitivity. When using any technique to increase sensitivity, it is important to pay attention to the fact that this technique does not complicate the process and does not stop LFA being a user-friendly test. 

The most common method to functionalize AuNPs with aptamers is through the affinity of -thiol to AuNPs, which is possible by using thiolated aptamers.

Following these limited studies, the use of aptamer-based LFAs has not yet been commercialized. Therefore, solutions should be provided for the further use of aptamers in LFA tests as well as their commercialization. One of these solutions is the development of aptamers for emerging analytes (e.g., SARS-CoV-2) and the design of rapid diagnostic tests including LFA strips and the subsequent commercialization of these tests. This strategy could pave the way for the development and commercialization of aptamer-based LFA strips for other analytes.

In summary, considering the many advantages of both aptamers and LFAs, their combination in the development of clinical diagnostic tests and their subsequent commercialization is strongly recommended.

## Figures and Tables

**Figure 2 pharmaceuticals-15-00090-f002:**
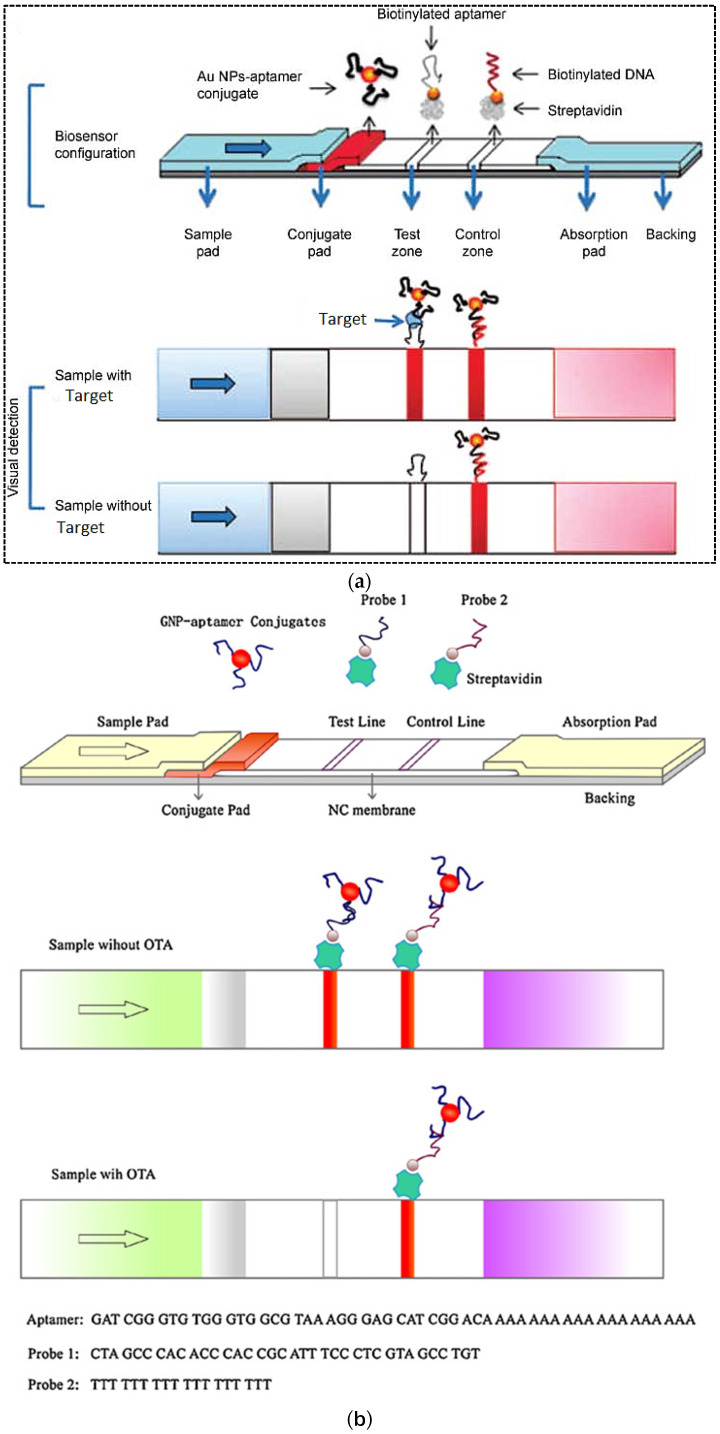
Principle of (**a**) sandwich aptamer-based LFA for the detection of thrombin; (**b**) competitive aptamer-based LFA for the detection of ochratoxin A. Reproduced with permission from [28,29], respectively.

**Figure 4 pharmaceuticals-15-00090-f004:**
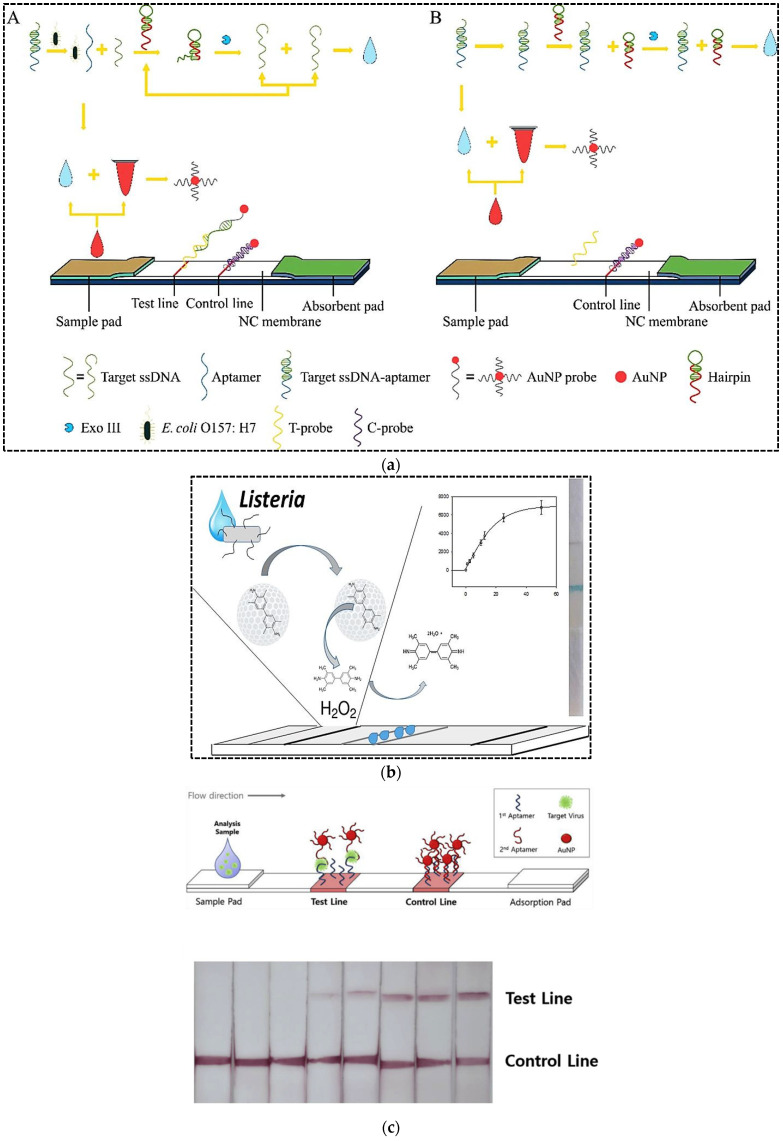
(**a**) Principle of aptamer-exonuclease III (Exo III)-assisted amplification-based LFA for the detection of *E. coli* O157:H7: (A) in the presence of target; (B) in the absence of target; (**b**) A LFA based on aptamer-gated silica nanoparticles and release of signal molecules for the detection of *Listeria monocytogenes*; (**c**) whole virus particle detection using a sandwich LFA. Reprinted with permission from [49,50,51], respectively.

**Figure 5 pharmaceuticals-15-00090-f005:**
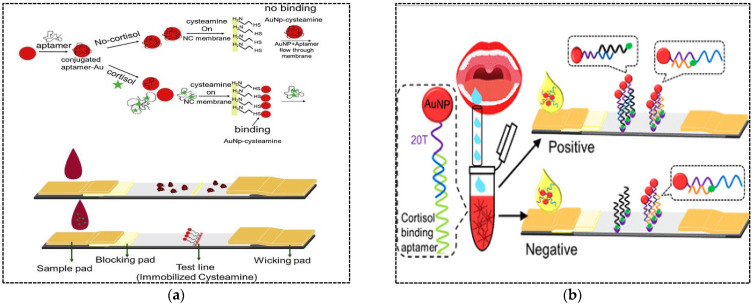
(**a**) A LFA based on adsorption–desorption of an aptamer on the surface of AuNPs for cortisol detection; (**b**) A LFA based on AuNPs conjugated with a duplex aptamer for the detection of cortisol in saliva. Reproduced from [56,57], respectively, with permission.

**Figure 6 pharmaceuticals-15-00090-f006:**
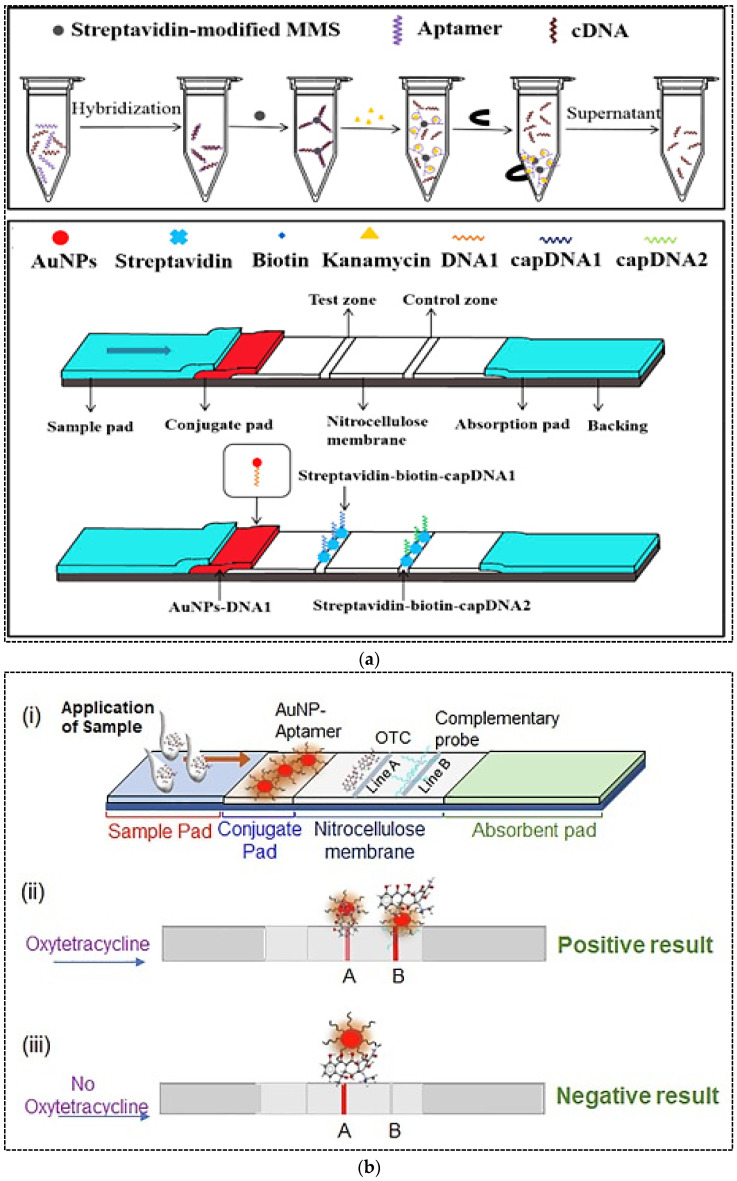
(**a**) Schematic representation of magnetic microsphere-assisted LFA for kanamycin detection; (**b**) A competitive aptamer-based LFA for the detection of oxytetracycline; (**c**) A competitive aptamer-based LFA for ampicillin detection. Reprinted from [62,63,64], respectively, with permission.

**Table 1 pharmaceuticals-15-00090-t001:** Aptamer-based lateral flow assays for detection of biomarkers.

Strategy	Biomarker	Label	Test Line	Control Line	LOD	Detection Range	Matrix	Ref.
Conjugation of AuNPs with dopamine duplex aptamer and dissociation of duplex in the presence of target	Dopamine	AuNPs	Streptavidin-biotinylated cDNA3	Streptavidin-biotinylated cDNA1	50 ng mL^−1^	nr *	Urine	[37]
Desorption of biotin-modified aptamer from AuNPs surface in the presence of analyte	HER2	AuNPs	Streptavidin and pullulan mixture	Cationic charged PDDA polymer	20 nM	nr	Human serum	[38]
Competitive reaction between CA125 conjugated with AuNPs and unlabeled CA125 for binding to capture probe	CA125	AuNPs nanozyme	CA125 specific aptamer	-	5.21 U mL^−1^	7.5–200 U mL^−1^	Human serum	[40]
An aptamer-based hook-effect-recognizable three-line LFA	Thrombin	AuNPs	A biotinylated aptamer with poly A tail (T-DNA)	A biotinylated poly T oligonucleotide probe (C-DNA)	0.85 nM	1 nM–100 µM	Human serum	[41]
Binding of a biotinylated aptamer to target in the sample and reaction with AuNPs-streptavidin conjugate, then subsequent capturing by the antibody at the test line	Osteopontin	AuNPs	Osteopontin antibody	Complementary ssDNA	0.1 ng mL^−1^	10–500 ng mL^−1^	Human serum	[42]
A sandwich LFA based on biotin-labeled primary aptamer immobilized on streptavidin coated membrane as a capturing probe and secondary aptamer conjugated with AuNPs as a signaling probe.	Vaspin	AuNPs	Streptavidin-biotinylated aptamer	Streptavidin-biotinylated complementary aptamer	0.105 nM	0.105–25 nM	Human serum	[43]
Aptamer-quantum dot conjugate towards thrombin, and antibody-quantum dot conjugate against interleukin-6	Thrombin Interleukin-6	Green and red quantum dots	Streptavidin-biotinylated aptamer against thrombin Streptavidin- interleukin antibody	Anti-mouse antibody	3 nM 100 pM	nr	Human serum	[44]
Visual multiple recognition of protein biomarkers based on an array of aptamer- modified AuNPs	Thrombin Mucin Carcinoembryonic antigen	AuNPs	T1: Streptavidin T2: streptavidin-biotinylated mucin 1 protein test DNA	Streptavidin-biotinylated thrombin control DNA	1.61 nM 1.13 nM 0.7 nM	3.2–250 nM 1.6–400 nM 0.8–300 nM	Human serum	[45]

* nr: not reported.

**Table 2 pharmaceuticals-15-00090-t002:** Aptamer-based lateral flow assays for detection of microbial analytes.

Strategy	Microbial Analyte	Label	Test Line	Control Line	LOD	Detection Range	Matrix	Ref.
Combining aptamer-exonuclease III-assisted amplification with LFA	*E. coli* O157:H7	AuNPs	Biotinylated poly T oligonucleotide probe	Biotinylated complementary sequence to the AuNPs probes	7.6 × 10^1^ cfu mL^−1^ 8.35 × 10^2^ cfu mL^−1^	10^2^–10^6^ cfu mL^−1^	Culture media Milk	[49]
Aptamer-gated release of TMB as signal molecule and HRP activity to generate signal	*L. monocytogenes*	Silica nanoparticles	Immobilized HRP	No control line	53 cells mL^−1^	nr	Chicken	[50]
Sandwich LFA for the detection of whole virus particles	Avian influenza H5N2	AuNPs	Biotin modified primary aptamer	Biotin modified poly A sequence	1.2 × 10^6^ EID_50_ mL^−1^	1.2 × 10^6^ EID_50_ mL^−1^–1 × 10^7^ EID_50_ mL^−1^	Duck’s feces	[51]
A dual recognition element LFA using both aptamer and antibody	Influenza virus (strain A/H3N2/Panama/2007/99)	AuNPs	Streptavidin	Antibody	2 × 10^6^ virus particle	-	-	[52]
Pre-enrichment with magnetic nanoparticles conjugated with aptamer–ssDNA1and detection of released ssDNA1using LFA strip and AuNPs capture probe	*Salmonella* Typhimurium	AuNPs	Streptavidin-biotin modified ssDNA complementary with ssDNA1	Streptavidin-biotin modified poly A ssDNA complementary with poly T ssDNA on AuNPs surface	4.1 × 10^2^ cfu mL^−1^	8.6 × 10^2^–8.6 × 10^7^ cfu mL^−1^	Milk	[54]
Complex formation between AuNPs-aptamer1 conjugate, analyte and apatmer 2 immobilized on the test line	*Salmonella* Typhimurium *E. coli* O157:H7 *Staphylococcus aureus*	AuNPs	Streptavidin-biotin modified aptamer2	Streptavidin-biotin modified complementary DNA with AuNPs-aptamer conjugate	5 × 10^3^ cfu mL^−1^ 3 × 10^4^ cfu mL^−1^ 2 × 10^4^ cfu mL^−1^	nr	Food samples	[55]

**Table 3 pharmaceuticals-15-00090-t003:** Aptamer-based lateral flow assays for detection of hormones.

Strategy	Hormone	Label	Test Line	Control Line	LOD	Detection Range	Matrix	Ref.
Desorption of specific aptamer from AuNPs surface in the presence of target molecule	Cortisol	AuNPs	Cysteamine	-	1 ng mL^−1^	nr	Sweat	[56]
Dissociation of specific aptamer from duplex probe conjugated with AuNPs in the presence of target	Cortisol	AuNPs	Biotin-modified oligonucleotide probe	Biotin-modified poly A probe	0.37 ng mL^−1^	0.5–15 ng mL^−1^	Salivary	[57]
Dissociation of specific aptamer from duplex probe conjugated with AuNPs in the presence of target	Progesterone	AuNPs	Streptavidin	-	5 nM	nr	Water	[58]

**Table 4 pharmaceuticals-15-00090-t004:** Aptamer-based lateral flow assays for detection of antibiotics.

Strategy	Antibiotic	Label	Test Line	Control Line	LOD	Detection Range	Matrix	Ref.
Magnetic sepration of analyte by aptamer-cDNA duplex conjugated with magnetic microspheres; then detection of cDNA by LFA strip	Kanamycin	AuNPs	Streptavidin-biotin-capture DNA1	Streptavidin-biotin-capture DNA2	4.96 nm	5–500 nM	Milk, milk products, honey	[62]
Competitive reaction between analyte in sample and analyte-career protein conjugate on the test line for binding to AuNPs-aptamer	Oxytetracycline	AuNPs	Oxytetracycline-carrier protein	Biotin-modified complementary probe	5 ng mL^−1^	nr	Milk	[63]
Competitive reaction between analyte in sample and C-reactive protein conjugated with biotin for binding to AuNPs-aptamer-mFc	Ampicillin	AuNPs	Streptavidin	α-mouse antibody	185 mg L^−1^	nr	Milk	[64]
Complex formation between analyte in sample, labeled DNA on conjugate pad and capture DNA on the test line	Ampicillin Kanamycin	Hexachloro-6-carboxyfluorescein	Biotin-modified ampicillin capture DNA Biotin-modified kanamycin capture DNA	-	0.06 ng L^−1^ 0.015 ng L^−1^	0.5–500 ng L^−1^ 0.5–1000 ng L^−1^	Water	[65]

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
