# Peer review of "Aptamer-Based Lateral Flow Assays: Current Trends in Clinical Diagnostic Rapid Tests"

_pharmaceuticals, 2022, doi:10.3390/ph15010090_

Round 1

Reviewer 1 Report

The authors of Aptamer-based Lateral Flow Assays: Current Trends in Clinical Diagnostic Rapid Tests aim to give a comprehensive overview of the current developments and applications of lateral flow tests. They give a decent overview of recent developments, but some aspects of current LFAs fall short. While they start out explaining the basic concepts with antibody-based assays, they quickly change over to only writing about aptamers. The authors completely ignore the current pandemic and the development of LFA for covid testing.

The manuscript starts with the first commercially available LFA and a few use cases. It proceeds with the design principles and general build of a common LFA. An extensive comparison between the sandwich and competitive format is given, followed by common antibody/aptamer labels. When describing the general build of an LFA, a high-quality illustration would be in order. Figure 1a does not suffice and is of poor quality. It also does not show how the pads and membranes of the assembly usually overlap and are not placed edge to edge.

Part 3 explains the replacement of antibodies with aptamers. The authors focus on aptamers in the further parts of the manuscript, which reflects the current developments. However, it is quite confusing that every basic aspect of LFAs is described with antibodies, only to be dropped in favor of aptamers. A review focused on LFAs overall should not neglect antibody based LFAs. Furthermore, (deoxy-)oligonucleotides are not only employed as aptamers but also to detect complementary sequences. It is suggested that the authors either focus on aptamer based LFAs or take a broader approach with their manuscript.

The manuscript proceeds with the clinical applications of aptamer based LFAs. Lines 260f give percentages for the usage of LFA strips. These percentages are not backed by literature or references. The given reference does not include the source for these numbers. It is strongly suggested that these numbers are corrected, backed by a source or removed. The same is true for lines 413ff.

Lines 277-282 suggest that aptamers are less susceptible to chemical environment and similar structure. This is not necessarily true and needs to be backed by references.

Lines 435-447 are written too complex and do not do a good job in explaining the principle. Figure 4b does not help with the explanation and does not even illustrate what is written in the paragraph. A rework of this part is suggested.

Since the authors dedicate one whole part to the detection of microbial analytes, one wonders why the current SARS-CoV2 pandemic is not mentioned. It has caused a massive increase in demand and use of LFAs. A review on current developments in LFAs should include this topic and give examples on SARS-CoV2 testing.

Current and future challenges of LFA development are not addressed and should be discussed.

The manuscript ends with a balanced and well written conclusion. References appear adequate, except the missing ones mentioned above. The overall quality of the manuscript is satisfactory.

Author Response

The authors of Aptamer-based Lateral Flow Assays: Current Trends in Clinical Diagnostic Rapid Tests aim to give a comprehensive overview of the current developments and applications of lateral flow tests. They give a decent overview of recent developments, but some aspects of current LFAs fall short. While they start out explaining the basic concepts with antibody-based assays, they quickly change over to only writing about aptamers. The authors completely ignore the current pandemic and the development of LFA for covid testing.

Response: Thank you very much for your valuable comments. In order to address these weaknesses and improving the structure of the article based on the opinions of the respected referee, some explanations were included in the text on lines 45-50, 55-57, and 113-118. 

The manuscript starts with the first commercially available LFA and a few use cases. It proceeds with the design principles and general build of a common LFA. An extensive comparison between the sandwich and competitive format is given, followed by common antibody/aptamer labels. When describing the general build of an LFA, a high-quality illustration would be in order. Figure 1a does not suffice and is of poor quality. It also does not show how the pads and membranes of the assembly usually overlap and are not placed edge to edge.

Response: Thanks for your good hint. A suitable figure (Fig. 1a) with good quality and a good view of the overlap of the membranes was replaced.

Part 3 explains the replacement of antibodies with aptamers. The authors focus on aptamers in the further parts of the manuscript, which reflects the current developments. However, it is quite confusing that every basic aspect of LFAs is described with antibodies, only to be dropped in favor of aptamers. A review focused on LFAs overall should not neglect antibody based LFAs. Furthermore, (deoxy-)oligonucleotides are not only employed as aptamers but also to detect complementary sequences. It is suggested that the authors either focus on aptamer based LFAs or take a broader approach with their manuscript.

Response: Dear referee, thanks for this valuable comment. However, in my opinion, it is necessary for the reader to be acquainted at the beginning of the article with the general principles of this method, which was established with antibodies at the beginning of development. For this reason, in this article, we have tried to present the content step by step and the reader gradually enters into the issue of replacing aptamer in the lateral flow method.

We must not forget that antibodies are still the most effective diagnostic elements in many methods and are currently being studied that may be able to replace antibodies in the future. Also, currently all commercial examples of lateral flow method are antibody based. Therefore, it is necessary to provide information about antibody-based lateral flow in this article. However, the replacement of these diagnostic elements with aptamers and principles of detection in aptamer-based lateral flow is fully explained (section 3) and was not merely a drop of aptamers in antibody-based detection principles.

Based on these explanations, I ask the esteemed referee, with all due respect, to agree with the present format of the article in this regard.

The manuscript proceeds with the clinical applications of aptamer based LFAs. Lines 260f give percentages for the usage of LFA strips. These percentages are not backed by literature or references. The given reference does not include the source for these numbers. It is strongly suggested that these numbers are corrected, backed by a source or removed. The same is true for lines 413ff.

Response: Thanks a lot for your precision. The reference of these percentages was included (line 219).

Regarding line 413 (Moreover, they need expensive instruments and they are not suitable for use in deprived areas ….), the related reference was included (line 360).

Lines 277-282 suggest that aptamers are less susceptible to chemical environment and similar structure. This is not necessarily true and needs to be backed by references.

Response: The concept of “antibody-based LFA, specificity and sensitivity can be affected by other chemicals with similar structures” has been mentioned in the following article:

Koczula, K. M., & Gallotta, A. (2016). Lateral flow assays. Essays in biochemistry, 60(1), 111–120. https://doi.org/10.1042/EBC20150012.

The reference was included in the text (line 236).

Regarding “less susceptibility of aptamer to chemical environment” I must say that there is no discussion about this issue throughout the article and exactly according to your opinion, aptamers are not necessarily more stable to chemical environment.

Lines 435-447 are written too complex and do not do a good job in explaining the principle. Figure 4b does not help with the explanation and does not even illustrate what is written in the paragraph. A rework of this part is suggested.

Response: Thanks for your nice comment. This paragraph was rewritten for more understanding (lines 384-398).

Since the authors dedicate one whole part to the detection of microbial analytes, one wonders why the current SARS-CoV2 pandemic is not mentioned. It has caused a massive increase in demand and use of LFAs. A review on current developments in LFAs should include this topic and give examples on SARS-CoV2 testing.

Response: Dear referee, your opinion is quite logical and appropriate. We didn’t mention SARS-CoV2 detection because approximately all LFA developed for SARS-CoV2 detection is based on antibody, while the article has been focused on clinical uses of aptamer-based LFA. However, due to the importance of this global problem, some explanation was included in this section (lines 438-457)

Current and future challenges of LFA development are not addressed and should be discussed.

Response: Regarding this comment, a section was included in the manuscript and some challenges were explained (Lines 638-662)

The manuscript ends with a balanced and well written conclusion. References appear adequate, except the missing ones mentioned above. The overall quality of the manuscript is satisfactory.

Thank you very much for your constructive comments and your positive opinion.

Reviewer 2 Report

Comments to the Authors

The review described by Marjan Majdinasab et al., Aptamer-based Lateral Flow Assays: Current Trends in Clinical Diagnostic Rapid Tests was written very well from suitable articles. They focused on different aptamer based LFAs for a variety of targets (Biomarkers, Microbial analytes, hormones and antibiotics) with various mechanisms. Overall, I think it could be accepted only after minor revisions.

Please see the comments below.

  • Line 84-91: Figure 1 (b) competitive format scheme 2 did not explain in this paragraph.
  • Line 260: and 261-typo error for 69%, 28% and 3%
  • Line 396: In figure 3 caption, part d did not describe.
  • Line 463: typo error for aptamer and authors need to include explanation about control line.
  • Line 475: In the complex of aptamer- virus-mAb, need to include AuNPs.
  • Line 533-536: Authors explained only the presence of the target but did not describe the absence of target.
  • Line 564: The mixture of complex did not include gold nanoparticles
  • Line 693,694 and 695: The font size was different.

Author Response

  • Line 84-91: Figure 1 (b) competitive format scheme 2 did not explain in this paragraph.

Response: Thank you very much for your precision. Second type of competitive format was explained (lines 92-104)

  • Line 260: and 261-typo error for 69%, 28% and 3%

Response: The revision was made for this error (lines 218-219).

  • Line 396: In figure 3 caption, part d did not describe.

Response: Thank you very much. Capture of Fig. 3d was included.

  • Line 463: typo error for aptamer and authors need to include explanation about control line.

Response: typo error was revised and some explanation about control line was included (Lines 417-419)

  • Line 475: In the complex of aptamer- virus-mAb, need to include AuNPs.

Response: thanks for your precision. AuNPs” was included (Line 430)

  • Line 533-536: Authors explained only the presence of the target but did not describe the absence of target.

Response: Some explanations regarding “absence of target” was included in the text (Lines 491-492)

  • Line 564: The mixture of complex did not include gold nanoparticles

Response: AuNPs was included in the complex (lines 518-519)

  • Line 693,694 and 695: The font size was different.

Response: Smaller font size is related to the figure caption.

Reviewer 3 Report

In this manuscript, the use of aptamers in the construction and performance of lateral flow assays is presented and discussed.

The authors have conducted extensive research on the advantages of using aptamers compared to the use of antibodies. The literature selected is in line with the requirements of a review article and covers all major areas regarding the importance of aptamers for the detection of complex biomolecules.

Suggestions for improvement:

  1. the number of illustrations is clearly too high and should be reduced. Rather, some figures should be moved to the appendix of the manuscript.

2.The tables should be presented in landscape format.

  1. A short chapter should be dedicated to the topic of sensitivity and specificity in the comparison of aptamers LFA versus antibodies LFA.

Author Response

  1. The number of illustrations is clearly too high and should be reduced. Rather, some figures should be moved to the appendix of the manuscript.

Response: Dear referee, thank you very much for your comment. Since illustrations and figures can help to better understand the detection mechanism, please (with all respects) allow them to exist in the same format in the text. Thanks a lot for if you agree.

  1. The tables should be presented in landscape format.

Response: Tables are in landscape format in the original version submitted to the journal

  1. A short chapter should be dedicated to the topic of sensitivity and specificity in the comparison of aptamers LFA versus antibodies LFA.

Response: In this regard, short section was included in the manuscript (linea 611-636).